# Intensity of Symptoms and Perception of Quality of Life on Admission to Palliative Care: Reality of a Portuguese Team

**DOI:** 10.3390/healthcare12151529

**Published:** 2024-08-01

**Authors:** Florbela Gonçalves, Margarida Gaudêncio, Ivo Cristiano Soares Paiva, Francisca Rego, Rui Nunes

**Affiliations:** 1Portuguese Institute of Oncology Francisco Gentil Coimbra, 3000-075 Coimbra, Portugal; 4196@ipocoimbra.min-saude.pt; 2Faculty of Medicine, University of Porto, 4200-319 Porto, Portugal; mfrego@med.up.pt (F.R.); ruinunes@med.up.pt (R.N.); 3Health Sciences Research Unit: Nursing, Nursing School of Coimbra, 3000-232 Coimbra, Portugal; ivopaiva@esenfc.pt

**Keywords:** medical oncology, palliative care, quality of life, outcome assessment, symptom assessment

## Abstract

Background: Palliative care (PC) corresponds to an approach that enhances the quality of life for patients facing life-threatening diseases, such as cancer, as well as for their families. There are various models for providing palliative care. Early referral to PC of patients with advanced cancer has a significant positive impact on their quality of life. However, the criteria for early referral still remain controversial. Objectives: To evaluate patients’ symptomatic intensity and perception of quality of life on admission to a PC unit and to analyze these two variables according to different models of approach (outpatient and inpatient care). Methods: A cross-sectional, descriptive, and correlational study was conducted with a sample of 60 patients sequentially admitted to a PC unit from palliative outpatient consultations or other inpatient services in a tertiary hospital dedicated to oncology care. The evaluation protocol included a sociodemographic and medical questionnaire, the Edmonton Symptom Assessment Scale (ESAS), and the Palliative Care Outcome Scale (POS) completed by patients within the first 24 h after admission. Results: The participants were mostly male (61.7%), with a median age of 72 years. The majority of patients (*n* = 32; 53.3%) were undergoing outpatient treatment, while the remaining individuals (*n* = 28; 46.7%) were transferred from other hospital services (inpatient care). In the outpatient care group, higher scores for fatigue and dyspnea were observed. Conversely, in the inpatient care group, higher scores were observed for pain, depression, and anxiety. There were significant differences between the two groups regarding the POS dimensions of meaning of life, self-feelings, and lost time. In the inpatient group, there was a longer time between diagnosis and referral to PC; however, it was also in the inpatient group that there was less time between PC referral and first PC evaluation, between PC referral and PC unit admission, and between PC referral and death. There were no significant correlations between referral times and ESAS/POS scores in the inpatient and outpatient groups. Conclusions: The patients admitted to the Palliative Care Unit presented a high symptom burden and changes in the perception of quality of life. However, there are no statistically significant differences between one model of approach in relation to the other. It was found that poorer symptom control and quality of life were associated with a shorter referral time for PC, because this was only initiated after curative care was suspended, particularly in our institutional context. Early referrals to the PC team are essential not only to relieve symptom-related distress but also to improve treatment outcomes and quality of life for people with cancer.

## 1. Introduction

The World Health Organization (WHO) defines palliative care (PC) as an approach that improves the quality of life of patients and their families who are facing problems associated with life-threatening illness through the early identification and treatment of pain, alongside other physical, psychosocial, and spiritual problems [1].

In Portugal, access to palliative care is considered to be a universal right regulated in the Basic Health Law approved in 2012 [2]. According to this law, palliative care “must respect the autonomy, will, individuality, the dignity of the person and the inviolability of human life” [2]. It was the law that defined the creation of PC teams at the local level in Portugal, organized into PC units, in-hospital PC support teams, and community PC support teams [2].

Wiencek et al. describe various models for providing palliative based on the reality experienced in the United States of America—Ambulatory Palliative Care Clinic, Home-Based Palliative Care, Inpatient Palliative Care Units, and Inpatient Consultation Services [3].

Although a number of benefits have been attributed to inpatient and outpatient care individually, the relative impact of one in relation to the other is not as well-described [4,5,6,7,8]. The two environments are characterized by marked differences in the process of care [4,5,6,7,8]. A recent study has shown that an outpatient palliative care model reduces hospital admissions, improves quality of life, and prolongs patient survival [4,9].

Traditionally, PC for cancer patients was initiated only after the completion of curative treatment was finished [10]. However, it has been proven that early referral of these patients not only enhances their quality of life but may also potentially improve survival rates simultaneously [4].

Early referral to PC for patients diagnosed with advanced cancer significantly improves their overall quality of life [11]. Although the definition of early referral has not been well-established, some randomized trials described the following criteria for early referral: 2 to 3 months after the diagnosis of advanced disease and Eastern Cooperative Oncology Group (ECOG) performance status of 2 or less [12]. Other authors described referral time to palliative care in relation to death: first PC referrals more than one year before the patient‘s death are classified as “early” (ER); referrals between 3 and 12 months before death are classified as “intermediate”; and referrals made less than 3 months before the patient’s death are classified as “late“ (LR) [13].

Referral criteria are still controversial [10]. Severe symptoms, poor performance status, comorbidities, and prognosis are among the most common criteria [10].

The difficulty of early referral of cancer patients to PC is related to the difficulty in establishing a correct prognosis [11]. In fact, oncologists readily admit the difficulty in estimating survival of 3, 6, or 12 months [11]. Other reasons implicated in the delay of referral to PC are the scarcity of time during consultations to discuss PC and failures in the assessment of psychological, spiritual, or social needs [12].

Late referral of patients to PC has an impact on symptomatic control and quality of life [14]. Several studies have shown that early initiation of PC significantly improves the quality of life of patients with advanced cancer if initiated concomitantly with standard oncology care [14]. Humphreys et al. concluded that late referrals may have a negative impact on health outcomes [15]. So, it is important to design and implement hospital policies that encourage early referral to palliative care, particularly for advanced cancer patients [15]. Patients referred to palliative care within the first week of hospital admission experienced significantly shorter lengths of stay and lower in-hospital mortality rates compared to those referred after 1 week following admission [15].

As already mentioned, one of the objectives of PC is to promote quality of life for patients and their families. However, the perception of quality of life is very subjective and presented with different concepts.

According to the WHO, quality of life could be defined as the individual’s perception of their position in life, based on values, culture, goals, expectations, and concerns [16].

Several scales are used to measure quality of life in the cancer and PC context, including the Eastern Cooperative Oncology Group Performance Status Scale (ECOG) and the QLQ-C30 designed by the European Organization for Research and Treatment of Cancer (EORTC) [17].

It is recognized that assessment of PC and cancer patients’ quality of life requires that the emphasis of the measurement is not only on the relief of pain and other symptoms [17]. It is also important to consider the resolution of emotional, social, psychological, and spiritual problems, as well as information and establishing good contacts and support with family members [17].

The Palliative Care Outcome Scale (POS) seeks to assess these dimensions, having been developed particularly for cancer and palliative patients [17,18].

The general aim of this study is to evaluate patients’ symptomatic intensity and perception of quality of life on admission to a Palliative Care Unit.

The specific aims of the study are the following:-To analyze patients’ symptomatic intensity and perception of quality of life upon PC unit admission, according to different models of approach (outpatient and inpatient care);-To evaluate the referral time to PC in the global sample and according to different models of approach (outpatient and inpatient care);-To correlate the referral time to PC with patients’ symptomatic intensity and perception of quality of life according to different models of approach (outpatient and inpatient care).

## 2. Materials and Methods

The authors conducted a cross-sectional descriptive and correlational study, carried out at an oncological tertiary hospital, after approval from the Ethics Committee of the institution where the study was conducted (opinion No. TI 17/2020).

### 2.1. Data Collection

Data collection respected the rules of the Helsinki Protocol [19] and the Oviedo Convention [20].

The data measurement tools were distributed individually, accompanied by information explaining the nature and objectives of the study to each participant. Additionally, an informed consent form was provided, ensuring the confidentiality of the data.

After obtaining authorization through informed consent, the data were collected using an evaluation protocol specifically designed for this purpose. The protocol included a sociodemographic and medical questionnaire (including age, gender, origin service before admission, types of cancer, and stage at diagnosis), an Edmonton Symptom Assessment Scale (ESAS), and a Palliative Care Outcome Scale (POS).

This protocol was applied in the first 24 h following admission to the Palliative Care Unit.

The Edmonton Symptom Assessment Scale (ESAS) is a multi-dimensional assessment tool used for self-reporting the intensity of symptoms in daily assessments of palliative care patients [21]. It is implemented and valid in several countries [22]. The ESAS consists of a numerical scale that evaluates the 10 most common symptoms in cancer patients—pain, fatigue, nausea, depression, anxiety, sleepiness, well-being, dyspnea, appetite, and one other optional symptom, and it uses a score from 0 (no intensity) to 10 (worst intensity) to measure suffering associated with both physical and psychological symptoms [22].

In this study, the authors used a validated subtype of the ESAS, the Global Distress Score (GDS), a sum of the first 9 physical and psychosocial ESAS symptoms [10]. The GDS was calculated and grouped into 3 cohorts based on previous work and clinical experience as follows: high (GDS ≥ 35), moderate (16 ≥ GDS > 34), and low (GDS < 15) [23]. The Palliative Care Outcome Scale (POS) was developed in the United Kingdom, and it is a multi-dimensional scale that can be used to assess the quality of life of patients in palliative care [24]. In fact, the POS was developed to measure quality of care and well-being in patients in palliative care, but it is also a good measure of quality of life because it addresses aspects related to physical, psychological, and spiritual symptoms, practical and emotional concerns, as well as psychosocial needs of the patient and family. It presents two versions, one “self” for the patient and the other “proxy” for the health professional, with identical and reliable answers [24]. We used the original version of “self” with 11 questions, which are scored on a 5-point Likert scale [24]. It addresses aspects related to physical, psychological, and spiritual symptoms, practical and emotional concerns, as well psychosocial needs of the patient [24]. The score can vary from 0 to 40 points, with the latter score being the one that represents the worst quality of life [24]. The questions present in this scale and used in the study were the following: “1—Over the past 3 days, have you been affected by pain; 2—Over the past 3 days, have other symptoms e.g., nausea, coughing or constipation seemed to be affecting how you feel; 3—Over the past 3 days, have you been feeling anxious or worried about your illness or treatment; 4—Over the past 3 days, have any of your family or friends been anxious or worried about you; 5—Over the past 3 days, how much information have you and your family or friends been given; 6—Over the past 3 days, have you been able to share how you are feeling with your family or friends; 7—Over the past 3 days, have you felt that life was worthwhile; 8—Over the past 3 days, have you felt good about yourself as person; 9—Over the past 3 days, how much time do you feel has been wasted on appointments relating to your healthcare, e.g., waiting around for transport or repeating tests; 10—Over the past 3 days, have any practical matters resulting from your illness, either financial or personal, been addressed.

Subsequently, the authors collected data from the clinical files of patients included in the study to determine referral/admission times (time between the diagnosis and PC referral; time between PC referral and first PC evaluation; time between PC referral and unit admission; time between diagnosis and PC unit admission; and time between PC referral and death).

### 2.2. Sample

A consecutive sample of 60 patients admitted to a Palliative Care Unit was included in this study. These patients had already been previously assessed by palliative care (on an outpatient basis or through the in-hospital palliative care team).

The inclusion criteria encompassed adult cancer patients (≥18 years old) admitted to the Palliative Care Unit between January and March 2021. The study encompassed cancer patients who willingly consented to participate and demonstrated an understanding of the study’s objectives. Those unwilling to participate, individuals with communication disabilities, or those unable to comprehend and/or provide consent were excluded.

### 2.3. Statistical Analysis

The variables in the entire sample were characterized using the most appropriate descriptive statistics available.

Categorical and qualitative variables were described using absolute and relative frequencies (N and %). Quantitative variables were characterized by the mean, quartiles, minimum, and maximum.

Quantitative and qualitative variables were compared with the Mann–Whitney test. Correlations between qualitative variables were performed using the Spearman correlation coefficient.

The database was organized using Microsoft Excel^®^ 2016 software. Statistical analysis was performed using the IBM SPSS^®^ Statistics program (version 25.0 for Windows^®^). The tests were performed at a significance level of 95% (*p* < 0.05).

## 3. Results

### 3.1. Sociodemographic and Clinical Characteristics and POS and GDS Scores of the Sample

The sample predominantly comprises males (*n* = 37; 61.7%), with a median age of 72 years (from 43 to 94 years).

Most of the patients (*n* = 32; 53.3%) were receiving outpatient treatment (in palliative care ambulatory), while the remaining patients (*n* = 28; 46.7%) were transferred from other services of the institution, where they were already being assessed by the in-hospital palliative care team (hospitalized patients). Concerning the type of tumor, most patients had digestive tumors (41.7%), followed by head and neck tumors (20%) (Table 1).

Most of the patients were at stage 4 of cancer at diagnosis (71.6%). However, it was observed that three patients were at stage I and eight at stage II. These patients had other advanced diseases rather than cancer, and the objective of treatment was not curative.

The POS score exhibits a median of 15 (from 7 to 25) (Table 1). On the other hand, the median Global Distress Score (GDS) is 43, with extreme values from 17 to 62 (Table 1). Most participants (*n* = 42; 70%) scored a GDS ≥ 35, which corresponds to a high symptom burden (Table 1).

### 3.2. Relationship of Different Palliative Care Approaches in Symptom Control and Perception of Quality of Life upon Admission to the PC Unit

The scores of the ESAS scale of the groups in outpatient and inpatient care are presented in Table 2. In this sample, the results of the ESAS scale were compared for each symptom of the groups in outpatient and inpatient care.

In the outpatient care group, a higher median ESAS score was noted for fatigue and dyspnea. On the other hand, in the inpatient care group, a higher median ESAS score was observed for pain, depression, and anxiety. For the remaining symptoms, the median ESAS scores are the same in both groups. However, in none of the cases were there any statistically significant differences.

Regarding the GDS, there is a median score of 42.5 in both groups, which corresponds to a high symptom burden. However, there were no statistical differences, either (Table 2).

The POS scale scores for both the outpatient and inpatient care groups can be found in Table 3. The item “*If any, what have been your main problems in the last 3 days?*” was not considered in this study. The scores’ distribution was statistically significant for the following items: meaning of life, self-feelings, and lost time (*p* < 0.05) (Table 3). The inpatient group exhibited higher median scores concerning meaning of life and self-feelings (*p* = 0.031 and 0.047, respectively). However, concerning the “lost time” item, the outpatient group displayed a higher median score (*p* = 0.033) (Table 3).

### 3.3. Relationship between PC Referral/Admission Times, Symptom Intensity, and Perception of Quality of Life

The sample showed a median time between diagnosis and referral to palliative care of 516.53 days (minimum 12 and maximum 2938 days), a median time between PC referral and first palliative evaluation of 10.6 days (ranging from a minimum of 0 to a maximum of 42 days), a median referral time between diagnosis and PC unit admission of 564.59 days (ranging from a minimum of 24 to a maximum of 2954 days), a median referral time between PC referral and unit admission of 50.68 days (ranging from a minimum of 0 to a maximum of 632 days), and a median referral time between PC referral and death of 36 days (ranging from a minimum of 3 to a maximum of 893 days) (Table 4).

In the inpatient group, there was a longer time between diagnosis and referral to palliative care (600 days) (Table 5). On the other hand, it was also in the inpatient group that there was less time between PC referral and first PC evaluation (2 days), between PC referral and PC unit admission (3 days), and between PC referral and death (31 days) (Table 5). All of these results were statistically significant (*p* < 0.05) (Table 5).

There were no statistically significant differences between the two groups in terms of time between diagnosis and the PC unit admission (*p* = 0.288) (Table 5).

Finally, the authors intend to evaluate the correlation between referral times in PC, the severity of symptoms, and perception of quality of life.

It was found that the total score of ESAS was negatively correlated with referral times in the sample and outpatient group (Table 6). In general, higher scores on ESAS (worse symptom control) were related to shorter referral time for palliative care, particularly in the sample and outpatient group.

On the other hand, in the inpatient group, some referral times presented a positive correlation with ESAS score, although without statistical significance (Table 6). Some of these times were the time between PC referral and first PC evaluation, the time between PC referral and unit admission, and the time between PC referral and death. In these correlations, it was observed that the longer time was related to severe symptom burden.

As we can see in Table 6, there was only a statistically significant correlation between the total ESAS score and time between diagnosis and PC unit admission in the sample (−0.277; *p* = 0.049). This correlation was negative, which means that the longer time between diagnosis and PC admission was related to less severe symptoms.

It was found that the total score of POS was negatively correlated with referral times in the sample and the outpatient group (Table 7). In general, higher scores on POS (worse perception of life) were related to shorter time between PC referral and first PC evaluation.

On the other hand, in the inpatient group, some referral times presented a positive correlation with the POS score. Some of these times were the time between the diagnosis and PC referral, the time between PC referral and unit admission, and the time between the PC referral and death. In these correlations, it was observed that the longer time was related to worse perception of life.

As we can see in Table 7, there was only a statistically significant correlation between total POS score and time between PC referral and first PC evaluation in the sample (−0.348, *p* = 0.01). This correlation was negative, which means that the longer time between PC referral and first evaluation was related to less severe perception of life. At last, there was no significant correlation with POS score and referral times in the inpatient and outpatient groups.

## 4. Discussion

In this sample, there was a predominance of males, with a median age of 72 years. The predominance of males in our sample can be attributed to two factors: the small number of patients with breast cancer (10%) and digestive (41.7%) and head/neck (20%) tumors as the most common in this population. Colorectal and gastric tumors are the main causes of death related to cancer, and they have a higher mortality rate in men [25]. On the other hand, males are more susceptible to head and neck cancers than females, which had an impact on the incidence [26].

The most prevalent types of tumors in this study were digestive cancer (41.7%) and head/neck cancer (20%). Only 10% of the patients referred to PC had lung cancer. Fairchild et al. described that 29.9% of palliative patients had lung and digestive cancer, while only 3.2% had head/neck cancer [27]. In the sample, it was found that the majority of patients were in stage IV at diagnosis, which means that they presented an advanced disease.

This study was performed in a Portuguese PC unit dedicated to oncological PC. The PC team cares for around 400–500 patients per year and offers several types of support: an inpatient consultation team, a PC unit, and outpatient consultation.

Patients admitted to the PC unit may either come from outpatient consultation or other medical wards of the hospital after an evaluation of the in-hospital PC team. On the date of admission to the PC unit, 32 patients (53.3%) were admitted from the outpatient clinic. The remaining patients (*n* = 28; 46.7%) were hospitalized in other wards of the hospital with the support of the in-hospital palliative care support team (inpatient care). Hochman et al. performed a study that compared the characteristics of inpatients and outpatients [28]. In their sample, the majority of patients were outpatient (*n* = 417; 66%) [28], which was similar to the current study.

One of the authors’ objectives was to analyze patients’ symptomatic intensity and perception of quality of life at PC unit admission in general and according to different models of approach (outpatient and inpatient care).

In the sample, the Global Distress Score (GDS) presented a median score of 43, with a minimum of 17 and a maximum of 62. Most participants scored a GDS ≥ 35 (n = 42; 70%), which corresponds to a high symptom burden. The ESAS scale is a pragmatic, patient-centered symptom assessment tool that is easy to administer, interpret, and report [29]. It can identify responsiveness, minimal clinically important differences, and clusters of symptoms [26]. The ESAS scale is a unidimensional score that assesses only symptom intensity [29].

On the other hand, the sample presented a POS median score of 15, with a minimum of 7 and a maximum of 25. The POS is an outcome scale that inherently encompasses the multidimensionality of quality of life [30]. So, in the sample, the patients seem to have a relatively satisfactory quality of life despite the high symptomatic lack of control, but individualized care is essential. These results could be attributed to the comprehensive nature of POS, which evaluates dimensions beyond the common symptoms in advanced disease [24]. The POS measures a range of aspects, including physical symptoms, psychological well-being, and emotional and spiritual needs, but also information and support requirements [24].

In the outpatient care group, a higher median ESAS score was noted for fatigue and dyspnea. On the other hand, in the inpatient care group, a higher median ESAS score was observed for pain, depression, and anxiety. For the remaining symptoms, the median ESAS scores are the same in both groups. However, in none of the cases were there any statistically significant differences. Hochman et al. developed a study regarding symptom burden in palliative care, focusing on both outpatient and inpatient settings [28]. Outpatients reported more severe fatigue, pain, and nausea [28]. The inpatient group reported more intensity in symptoms, such as anorexia and dysphagia [28]. The differences between the two studies could be justified by the sample used. Our study focused specifically on oncological palliative patients, while Hochman et al.’s study encompassed both oncological and non-oncological patients. The intensity of symptoms in most cancer patients varies based on the type of cancer, its stage, and the treatments received [31,32]. In metastatic cancer, 35% to 96% of patients experience pain, 32% to 90% experience fatigue, and 10% to 70% experience shortness of breath [33].

Statistically significant differences were observed between the outpatient and inpatient care groups in terms of POS dimensions, such as meaning of life, self-feelings, and lost time. In this study, the distribution of values was statistically significant for meaning of life (*p* = 0.031) and self-feelings (*p* = 0.047), with a higher median score in the inpatient group, while lost time (*p* = 0.033) had a higher median score in the outpatient group. Hence, patients in the inpatient care group reported feeling that life was worthwhile and had a more positive self-perception compared to those in the outpatient group. Conversely, patients in the outpatient group expressed feelings of wasting time during healthcare appointments, such as waiting for transport or repeating tests.

Santos et al. carried out a study with the aim of identifying the factors that influence patients’ quality of life (QoL) in home-based palliative care [34]. In the study, 22.2% of patients felt good about themselves “all the time” and 55.5% of patients spent up to half of the day on health-related commitments, including self-care activities as well as others related to therapeutic prescriptions [34]. The observed differences can be attributed to the care settings—our patients were under the care of both outpatient and inpatient services, rather than receiving specialized home care teams. The inpatient group presented higher values of quality of life and care, which can be justified by the intensive psychological intervention performed by the in-hospital palliative care team. As expected, patients from outpatient care demonstrated higher scores in lost time due to consultations and their associated logistics.

Another objective of the study was to evaluate the referral time to PC in the sample and according to different models of approach (outpatient and inpatient care) in order to understand if the PC referral was early or late in our context. The sample exhibited a median time between diagnosis and referral to palliative care of 323 days, a median time between PC referral and the first PC evaluation of 3 days, a median referral time between diagnosis and PC unit admission of 439 days, a median referral time between PC referral and unit admission of 8 days, and a median referral time between PC referral and death of 36 days. All measured times had statistically significant differences between outpatient and inpatient care groups, except for the time between diagnosis and PC unit admission.

In this study, the median time between diagnosis and referral to palliative care was 51,653 days, which corresponds to more than 1 year. Taking into account that 71.6% of patients under study presented stage IV tumors at diagnosis, this time does not seem at all appropriate. Ferrell et al. described that all patients with advanced cancer should be followed up simultaneously by a multidisciplinary team with oncologist and palliative professionals in the first 8 weeks after diagnosis [35]. In our hospital context, institutional policy determined that referral for palliative care should preferably be carried out when curative care is suspended. The lack of health professionals dedicated to PC is one of the reasons for this option.

Furthermore, the time between diagnosis and PC referral was higher in the inpatient care group (600 days). These patients were often receiving systemic treatment with unrealistic expectations about the curative potential of chemotherapy and were referred to palliative care towards the end of their lives.

The median time between PC referral and first PC evaluation was 3 days in the sample, suggesting a quick PC response. This time was longer in the outpatient group (18 days) than the inpatient care group (2 days). Outpatient care is often the result of early referral during the course of the disease, allowing a longitudinal relationship to be built with the patient and family [36]. On the other hand, inpatient palliative care teams are often involved later in the course of the disease, usually providing management of acute symptoms and decision making in the context of hospitalization [36]. Capelas et al. presented a Delphi study that tried to define some quality indicators for palliative care teams [37]. One of the indicators defined was the initial assessment taking place within the first 48 h after referral to the PC team [37].

The median time between PC referral and unit admission was 8 days. This time was longer in the outpatient care group (88 vs. 3 days). The same is observed when we analyze the time between the PC referral and death.

The PRISMA study showed that, in Portugal, more than half of people would prefer to die at home, but, in reality, more than 60% of deaths occur in hospitals and only 20% at home [38]. Sousa et al. performed a study in a specialized Palliative Cancer Care Unit in Portugal that revealed that patients were admitted to the hospital from the emergency department (56.1%), followed by in-hospital transfer from oncology units (22%) [39].

Most of the patients followed by the in-patient PC team were in oncology wards. These patients were often already in supportive care, with high-burden symptoms, when they were referred to the PC team. This abrupt transition from curative treatments to palliative care can be traumatic for the patient and family, and it can cause hopelessness and fear of abandonment [40].

Tagami et al. conducted a study investigating the timing of palliative care referral for patients with advanced cancer [41]. It was reported that 20.9% (*n* = 102) of patients were referred at diagnosis, 11.4% (*n* = 56) during anti-cancer treatment, and 36.2% (*n* = 177) when there were no further treatment options [41]. An additional 5.1% reported that they had never been referred to specialized palliative care (SPC) during their lifetime [41]. The lack of symptomatic control and the suffering caused by the disease are best understood by a PC team that can accompany the patient both during standard oncology treatments and at the end of life [42].

So, the intervals between referral to palliative care, admission to the unit, and the terminal event are quicker in the inpatient group compared with the outpatient care group. In our sample, the PC team was effectively managing patients in an outpatient context for almost three months (88 days) before they needed hospitalization (either for clinical or psychosocial reasons like caregiver exhaustion), considering the late transition to palliative care [43].

The time between PC referral and death was about 36 days. The referral time to PC can range from the last 6 months to a year of life or just a few weeks. Definitions of early integration in PC may be different and controversial. The literature suggests that it takes at least 3 to 4 months to fully benefit from palliative care provided by a multidisciplinary team [44]. In 2018, the Lancet Commission on Palliative Care and Pain Relief concluded that there was a global shortage of palliative care and pain relief [45].

At last, the authors intend to correlate the referral times to PC with patients’ symptomatic intensity and perception of quality of life, according to different models of approach (outpatient and inpatient care).

It was found that the total score of ESAS was negatively correlated with referral times in the sample and outpatient group. In general, higher scores on ESAS (worse symptom control) were related to shorter referral time for palliative care. In the sample, there was a statistically significant correlation between the time of diagnosis and admission to the PC unit. A negative correlation was observed, which means that the shorter the time, the greater the intensity of symptoms upon admission. These findings are in line with the literature, in which poor symptom control is one of the main reasons for hospitalization in palliative care [46]. As the disease progresses, it is natural for physical symptoms to worsen.

On the other hand, in the inpatient group, some referral times presented a positive correlation with the ESAS score, although without statistical significance. In these correlations, it was observed that the longer time was related to severe symptom burden.

The same results were found in relation to the POS score. It was found that the total score of POS was negatively correlated with referral times in the sample and outpatient group. In general, higher scores on POS (worse perception of life) were related to shorter time between PC referral and first PC evaluation. There was only a statistically significant correlation between total POS score and time between PC referral and first PC evaluation in the sample (−0.348, *p* = 0.01). This correlation was negative, which means that the longer time between PC referral and first evaluation was related to less severe perception of life.

Early referrals to PC teams are essential not only to relieve symptom distress but also to improve treatment outcomes and quality of life for people with cancer [47]. Referrals to palliative care often come too late to significantly improve the quality of life of patients with cancer. Patients are typically referred to the palliative care team late in the course of their illness, with an average of 30–60 days before death [48]. Physicians face several barriers to early referral of patients to palliative care [49]. Among these barriers are the uncertainty regarding the evolution of the disease process and prognosis, the existence of periods of remission, the difficulty in transmitting bad news, the lack of knowledge about palliative care, the lack of time in consultations, and the difficulty of accessing this care [49].

Poorer symptom control and lower quality of life were observed in association with a shorter time between diagnosis and referral to palliative care, which can be explained by the late diagnosis of these patients, who in most cases were stage IV at presentation. Based on these results, an early referral of patients to palliative care is recommended. Most studies have found that better management of symptoms in patients with metastatic cancer is associated with increased survival [50,51]. PC could play a fundamental role in early interventions regarding symptom control as well as offering psychosocial support, contributing to improved survival rates and enhancing the quality of life of cancer patients [40]. An earlier referral to the PC team is associated with a reduced need for intensive medical care and improved quality outcomes for cancer patients [50,51].

Many authors have acknowledged the absence of consensus in the literature regarding patients who should be referred in the ambulatory setting [52]. Criteria, such as cancer diagnosis, prognosis, physical symptoms, performance status, psychosocial distress, and end-of-life care planning, are suggested as potential reasons for PC referral [52].

With this study, the authors have tried to understand the reality of PC in their institution, with a view to improving practices related to patient referral. The result of this study led to an institutional change seeking to refer hospitalized patients earlier in order to allow better PC intervention with patients and families.

### Limitations

The authors acknowledge certain limitations in the current research.

One of the limitations identified in this study is the relatively small sample size. On the other hand, the fact that this study took place only in one institution limited the sample size. The authors recognize this sample size limitation. However, our objective is not to extrapolate the results to other centers nationwide. In the near future, it would be interesting to extend this study to other national units. One of the objectives of this work is to draw attention, despite being a local study, to a reality that is observed in the PC area, even with existing knowledge and training on the topic.

In addition to the sample size, the authors recognize that the study covers a three-month period in 2021, which may seem like a short period of time.

Another limitation of the study is that the evaluation of these instruments was carried out only upon admission, with no other time to see the more specific effect of the palliative care intervention. The majority of patients admitted to the Unit have uncontrolled symptoms, which in itself constitutes a reason for hospitalization.

## 5. Conclusions

This study evaluated the symptomatic intensity and perception of quality of life of patients admitted to a Palliative Care Unit in a hospital dedicated to cancer care, but also referral times and the effect of different PC models of approach.

It was found that poorer symptom control and quality of life were associated with a shorter referral time for palliative care, because palliative care was initiated only after curative care was suspended, particularly in our institutional context.

In general, the authors observed that patients admitted to the Palliative Care Unit have a high symptom burden and changes in the perception of quality of life. However, there are no statistically significant differences between one model of approach in relation to the other.

The optimization of symptomatic intensity in PC is associated with improved quality of life, greater compliance, and even longer survival without prejudice to quality of life [34]. Symptoms may be due to cancer (direct or indirect consequences of cancer), treatment-related adverse effects, and/or comorbidities [53]. The symptom and quality of life measurement tools routinely used in clinical practice are, respectively, the Edmonton Symptom Assessment Scale (ESAS) and the Palliative Care Outcomes Scale (POS) [54,55,56].

Expanding this study to include the population of non-oncological palliative patients would be beneficial in future research. In addition, the use of alternative instruments to assess symptoms and quality of life, along with the exploration of additional reference criteria, could enrich future studies. More work is needed to define the most appropriate assessment tools for routine screening and referral to inpatient and outpatient palliative care. This approach would allow us to more comprehensively measure the impact of our work on patient referral and quality of life in palliative care, both in oncological and non-oncological contexts.

## Figures and Tables

**Table 1 healthcare-12-01529-t001:** Sociodemographic and clinical characteristics and POS score and GDS score of the sample (*n* = 60).

Variables	Descriptive
Gender, *n* (%)	
Male	37 (61.7)
Female	23 (38.3)
Age (years), median [Q1; Q3], min–max	72 [58.25; 79], 43–94
Origin service before admission, *n* (%)	
Ambulatory/outpatient	32 (53.3)
Hospitalization/inpatient	28 (46.7)
Types of cancer, *n* (%)	
Head and neck	12 (20)
Cutaneous	3 (5)
Digestive	25 (41.7)
Gynecological	5 (8.3)
Hematologic	1 (1.7)
Breast	6 (10)
Prostate	2 (3.3)
Lung	6 (10)
Stages of cancer, *n* (%)	
Stage I	3 (5)
Stage II	8 (13.3)
Stage III	6 (10)
Stage IV	43 (71.6)
POS score *, median [Q1; Q3], min–max	15 [13; 19], 7–25
ESAS score (GDS) *, median [Q1; Q3], min–max	43 [35; 48], 17–62
ESAS score (GDS) cohorts based on the total sum of the answers, *n* (%)	
Low (0–15)	0
Moderate (16–34)	13 (21.7)
High (≥35)	42 (70)

Q1—first quartile; Q3—third quartile; * total score was obtained by adding the items. ESAS—Edmonton Symptom Assessment Scale; POS—Palliative Outcome Scale; GDS—Global Distress Score.

**Table 2 healthcare-12-01529-t002:** Frequencies of the patients’ answers to the ESAS scale—inpatient vs. outpatient care (median ([Q1; Q3]).

	Outpatient Care (*n* = 34)	Inpatient Care (*n* = 26)	Mann–Whitney’s *p*-Values
Pain	2.5 [0; 7]	4.5 [0; 6.75]	0.875
Fatigue	7 [5; 7.25]	6.5 [5; 7]	0.835
Nausea *	0 [0; 6]	0 [0; 4.75]	0.789
Depression	5 [3.75; 6]	6 [4; 6.75]	0.219
Anxiety *	5 [5; 6]	6 [5.25; 7]	0.066
Drowsiness **	5 [1.75; 5]	5 [3.5; 6]	0.196
Appetite	6 [5; 7]	6 [5; 8]	0.866
Dyspnea	3 [0; 5.25]	0 [0; 5]	0.451
Well-being	6 [5; 7]	6 [5; 7]	0.076
Total score ***	42.5 [33; 49]	42.5 [36; 47]	0.850

* 1 missing; ** 3 missing; *** Global Distress Score (GDS).

**Table 3 healthcare-12-01529-t003:** Frequencies of the patients’ answers to the first 10 questions of the POS scale—inpatient vs. outpatient care (median ([Q1; Q3]).

Items\Score	Outpatient Care (*n* = 34)	Inpatient Care (*n* = 26)	Mann–Whitney’s *p*-Values
1. Pain	2 [0; 3]	2 [0; 3]	0.876
2. Other symptoms	2 [1.75; 3]	3 [2; 3]	0.110
3. Anxiety about disease	3 [2; 3]	3 [2; 3]	0.145
4. Family’s anxiety *	3 [2; 3]	3 [3; 3]	0.456
5. Information *	1 [1; 2]	2 [1; 2]	0.396
6. Sharing Information *	1 [1; 2]	2 [1; 2]	0.330
7. Meaning of life	1 [1; 1]	1 [1; 2]	0.031 **
8. Self-feelings	1 [1; 1]	1 [1; 2]	0.047 **
9. Lost time	0 [0; 0.5]	0 [0; 0]	0.033 **
10. Practical problems	0 [0; 2]	0 [0; 1.5]	0.628
Total Score	15 [12; 18]	17 [14; 20]	0.041

* 1 missing; ** *p* < 0.05.

**Table 4 healthcare-12-01529-t004:** Times related to palliative care (admission, evaluation, referral, and death) of the sample (*n* = 60).

Variables	Descriptive
Time between the diagnosis and PC referral, median [Q1; Q3], min–max (days) *	323 [196.5; 651], 12–2938
Time between PC referral and first PC evaluation, median [Q1; Q3], min–max (days) **	3 [1; 22.75], 0–42
Time between diagnosis and PC unit admission, median [Q1; Q3], min–max (days) ***	439 [221.75; 682.75], 24–2954
Time between PC referral and unit admission, median [Q1; Q3], min–max (days) ****	8 [2; 47], 0–632
Time between the PC referral and death, median [Q1; Q3], min–max (days) *****	36 [11; 104], 3–893

Q1—first quartile; Q3—third quartile; PC—palliative care; *—3 missing; **—4 missing; ***—4 missing; ****—4 missing; *****—7 missing.

**Table 5 healthcare-12-01529-t005:** Times related to palliative care (admission, evaluation, referral, and death)—inpatient vs. outpatient care (median).

	Outpatient Care (*n* = 32)	Inpatient Care (*n* = 28)	Mann–Whitney’s *p*-Values
Time between the diagnosis and PC referral (days)	446	600	0.044 *
Time between PC referral and first PC evaluation (days)	18	2	<0.001 **
Time between PC referral and unit admission (days)	88	3	<0.001 **
Time between diagnosis and PC unit admission (days)	535	602	0.288
Time between the PC referral and death (days)	133	31	<0.001 **

* *p* < 0.05; ** *p* < 0.01.

**Table 6 healthcare-12-01529-t006:** Spearman’s correlations (r, *p*-value) between referral times and the ESAS total score in the sample and in the groups—inpatient vs. outpatient care.

	All Patients (*n* = 60)	Outpatient Group (*n* = 32)	Inpatient Group (*n* = 28)
Time between the diagnosis and PC referral	−0.168 (0.235)	−0.175 (0.373)	−0.233 (0.273)
Time between PC referral and first PC evaluation	−0.141 (0.323)	−0.198 (0.313)	0.043 (0.846)
Time between PC referral and unit admission	−0.093 (0.517)	−0.251 0.198	0.322 (0.134)
Time between diagnosis and PC unit admission	−0.277 (0.049) *	−0.271 (0.162)	−0.322 (0.134)
Time between the PC referral and death	−0.214 (0.139)	−0.359 (0.071)	0.053 (0.811)

* *p* < 0.05.

**Table 7 healthcare-12-01529-t007:** Spearman’s correlations (r, *p*-value) between referral times and the POS total score in the sample and in the groups—inpatient vs. outpatient care.

	All Patients (*n* = 60)	Outpatient Group (*n* = 32)	Inpatient Group (*n* = 28)
Time between the diagnosis and PC referral	0.083 (0.548)	−0.078 (0.686)	0.108 (0.601)
Time between PC referral and first PC evaluation	−0.348 (0.01) **	−0.322 (0.089)	−0.034 (0.873)
Time between PC referral and unit admission	−0.240 (0.081)	−0.225 (0.240)	0.164 (0.434)
Time between diagnosis and PC unit admission	−0.005 (0.971)	−0.124 (0.523)	0.096 (0.650)
Time between the PC referral and death	−0.263 (0.062)	−0.169 (0.410)	−0.049 (0.817)

** *p* < 0.01.

## Data Availability

The datasets generated and analyzed during the current study are available from the corresponding author upon reasonable request.

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
