# Peer review of "Intensity of Symptoms and Perception of Quality of Life on Admission to Palliative Care: Reality of a Portuguese Team"

_healthcare, 2024, doi:10.3390/healthcare12151529_

Round 1
Reviewer 1 Report
Comments and Suggestions for Authors
Line 79 - rather than stating “several oncologists” would read better stating something like “ oncologists readily admit the difficulty in estimating survival of….”
Table 5 - unit of time should list Days
I am an outpatient Palliative physician so my perspective comes from that angle. I wonder what the outpatient palliative team is composed of, how many clinicians, compared to the inpatient team. In the US there are usually more inpatient clinicians than outpatient, leading to longer time to consultation.
It is not surprising to see that pain was not as significant in those who were seen by palliative care as outpatients.
Well written paper, great job.
Author Response
Reviewer 1
Comments 1: Line 79 - rather than stating “several oncologists” would read better stating something like “ oncologists readily admit the difficulty in estimating survival of….”
Response 1: Thank you for poiting this suggestion. We already changed in the article.
Comments 2: Table 5 - unit of time should list Days
Response 2: Thank you for poiting this suggestion. We already changed in the article.
Comments 3: I am an outpatient Palliative physician so my perspective comes from that angle. I wonder what the outpatient palliative team is composed of, how many clinicians, compared to the inpatient team. In the US there are usually more inpatient clinicians than outpatient, leading to longer time to consultation.
It is not surprising to see that pain was not as significant in those who were seen by palliative care as outpatients.
Response 3: Thank you for pointing this out. We agree with this comment. However, we do not considered the number of physicians relevant for the study. In fact, the number of inpatient and outpatient palliative physicians in our team is equal.
Reviewer 2 Report
Comments and Suggestions for Authors
Congratulations to this intersting manuscript! I would like to have a short explanation why there 3 patients with stage I and 8 patients with stage II disease. Were they in a curative setting?
Additionally, the manuscript would benefit from a more detailed discussion on the implications of the findings for clinical practice and policy.
Clarifying the criteria for early referral and addressing potential biases in patient selection could strengthen the study's conclusions.
Good luck with your manuscript!
Author Response
Reviewer 2
Comments 1: I would like to have a short explanation why there 3 patients with stage I and 8 patients with stage II disease. Were they in a curative setting?
Response 1: Thank you for poiting this suggestion. We already changed in the article and explaines better with result.
Comments 2: Additionally, the manuscript would benefit from a more detailed discussion on the implications of the findings for clinical practice and policy.
Response 2: Thank you for poiting this suggestion. We already changes in the article. With this study, the authors try to understand the reality of PC in their institution, with a view to improving practices related to patient referral. The result of this study led to an institutional change, seeking to refer hospitalized patients earlier, in order to allow better PC intervention with patients and families.
Comments 3: Clarifying the criteria for early referral and addressing potential biases in patient selection could strengthen the study's conclusions.
Response 3: Throughout our article, we discuss the issue of early referral. There is no consensus in the literature regarding this topic. Most studies recommend an average of 3-6 months before death. In practice, the authors consider that referral in the last days of life, as observed in the study, is not suitable for an effective intervention by Palliative Care team.
Reviewer 3 Report
Comments and Suggestions for Authors
The article addresses an important topic discussed in the context of palliative care. The text is well-written but contains several weaknesses. The last point listed below is the most significant flaw of the article:
(1).The authors use one of the key concepts in medical and ethical debates, ‘quality of life’. This term is ambiguous. There are disputes concerning this term in the literature. In the article, it is necessary to specify what the authors mean when they refer to this term. In other words, the term ‘quality of life’ needs to be defined and characterized. This should be done in the Introduction.
(2). Lines 376-377 – The decisions to discontinue therapy, mentioned by the authors, are referred to as the abandonment of medically futile therapy. Thus, the physician (medical team) abandons futile therapy in favor of palliative care. There is a broad discussion on this topic in the literature.
(3). The article should clearly separate the Discussion from the Limitations. In other words, a separate Limitations section should be introduced. Moreover, a limitation that the authors do not mention is that the presented studies cover a three-month period in 2021.
(4). The biggest weakness of the article is the local nature of the presented studies and the small number of patients studied. The authors themselves agree with this opinion. Hence, it is difficult to unequivocally state whether the presented research results can be extrapolated to other centers in the country. Furthermore, the local nature of the studies contributes little to the international discussion to which the journal Healthcare is dedicated.
Author Response
Reviewer 3
Comments 1: The authors use one of the key concepts in medical and ethical debates, ‘quality of life’. This term is ambiguous. There are disputes concerning this term in the literature. In the article, it is necessary to specify what the authors mean when they refer to this term. In other words, the term ‘quality of life’ needs to be defined and characterized. This should be done in the Introduction.
Response 1: Thank you for poiting this suggestion. We already changes in the article.
Comments 2: Lines 376-377 – The decisions to discontinue therapy, mentioned by the authors, are referred to as the abandonment of medically futile therapy. Thus, the physician (medical team) abandons futile therapy in favor of palliative care. There is a broad discussion on this topic in the literature.
Response 2: Thank you for poiting this suggestion. We already changes in the article and decided to eliminate this topic, because this discussion is not the objective of the study.
Comments 3: The article should clearly separate the Discussion from the Limitations. In other words, a separate Limitations section should be introduced. Moreover, a limitation that the authors do not mention is that the presented studies cover a three-month period in 2021.
Response 3: Thank you for poiting this suggestion. We already changes in the article.
Comments 4: The biggest weakness of the article is the local nature of the presented studies and the small number of patients studied. The authors themselves agree with this opinion. Hence, it is difficult to unequivocally state whether the presented research results can be extrapolated to other centers in the country. Furthermore, the local nature of the studies contributes little to the international discussion to which the journal Healthcare is dedicated.
Response 4: Thank you for pointing this suggestion. We already answer to this topic in the article.
The authors recognize this weakness as a limitation to the work. However, our objective is not to extrapolate the results to other centers nationwide, although we know that this reality is felt by other colleagues who dedicate themselves to the area. One of the main objective of this work is to draw attention, despite being a local study, to a reality that is observed in the area of Palliative Care, even with existing training on the topic.
Round 2
Reviewer 3 Report
Comments and Suggestions for Authors
I have no more comments.